# On the Aggregation and Sensing Properties of Zinc(II) Schiff-Base Complexes of Salen-Type Ligands

**DOI:** 10.3390/molecules24132514

**Published:** 2019-07-09

**Authors:** Giuseppe Consiglio, Ivan Pietro Oliveri, Salvatore Failla, Santo Di Bella

**Affiliations:** Dipartimento di Scienze Chimiche, Università di Catania, I-95125 Catania, Italy

**Keywords:** Schiff-bases, salen, zinc(II) complexes, molecular aggregation, sensing

## Abstract

The zinc(II) ion forms stable complexes with a wide variety of ligands, but those related to Schiff-bases are among the most largely investigated. This review deals with the peculiar aggregation characteristics of Zn(II) Schiff-base complexes from tetradentate N_2_O_2_ salen-type ligands, L, derivatives from salicylaldehydes and 1,2-diamines, and is mostly focused on their spectroscopic properties in solution. Thanks to their Lewis acidic character, ZnL complexes show interesting structural, nanostructural, and aggregation/deaggregation properties in relation to the absence/presence of a Lewis base. Deaggregation of these complexes is accompanied by relevant changes of their spectroscopic properties that can appropriately be exploited for sensing Lewis bases. Thus, ZnL complexes have been investigated as chromogenic and fluorogenic chemosensors of charged and neutral Lewis bases, including cell imaging, and have shown to be selective and sensitive to the Lewis basicity of the involved species. From these studies emerges that these popular, Lewis acidic bis(salicylaldiminato)Zn(II) Schiff-base complexes represent classical coordination compounds for modern applications.

## 1. Introduction

Molecular aggregation is a topic of current interest [1] playing a crucial role in the development of new materials [2,3,4,5] including mimicking biological systems [6,7], with different spectroscopic, electrical, or magnetic properties with respect to those of component molecules.

Schiff-base derivatives from substituted salicylaldehydes and amines are suitable template ligands for many metal ions. Zn(II) complexes are among those mostly investigated. Because of the d^10^ electron configuration of the Zn(II) ion, it is not subjected to ligand field stabilization, and forms stable complexes with various Schiff-base ligands. Among them, bis(salicylaldiminato) Zn(II) complexes of tetradentate salen-type ligands, L, derivatives from salicylaldehydes and 1,2-diamines, are attracting interest for their photophysical and catalytic properties [8,9,10,11,12,13].

Further interest is drawn by their aggregation properties. Actually, ZnL complexes are versatile synthons to supramolecular architectures [14,15,16]. This is because these complexes are Lewis acidic species which are stabilized through the axial coordination of a donor, thus saturating their coordination sphere, with formation of adducts or aggregate species having penta-coordinated, distorted square-pyramidal structures [15]. As the aggregation take place through the coordination to Zn(II) atom, this allows for a different control and fine tuning of the resulting self-assembled architecture. Therefore, a variety of molecular aggregates and supramolecular assemblies were achieved [10,11,12,13,14,15,16]. Another interesting aspect related to the Lewis acidic character of these ZnL complexes is their potential application as sensors of Lewis bases, because of their spectroscopic changes owing to the formation of monomeric adducts [11,17,18].

The usefulness of these ZnL complexes is thus manifold, as they offer a variety of fundamental and applicative research studies.

In this review, we report on the aggregation properties of ZnL Schiff-base complexes (Figure 1), mostly focused on their spectroscopic properties in solution varying the coordinating properties of the solvent, and on the relevant spectral changes as a consequence of deaggregation. The easy formation of adducts rends these ZnL complexes useful sensors of Lewis bases. The review will be not exhaustive and essentially covers the research activity of the last decade on this topic.

## 2. Lewis Acidic Properties of ZnL Complexes and Related Structures

The Lewis acidic character of ZnL complexes can be related to the structure of the tetradentate N_2_O_2_ Schiff-base ligand. In fact, the ring strain that would be generated by the diimine bridge does not allow to reach a tetrahedral coordination around the Zn(II) atom, constraining the metal center in a distorted planar geometry (Figure 2). As a consequence, monomeric ZnL complexes are unstable, coordinatively unsaturated species. Conversely, ZnL′_2_ complexes of bidentate NO Schiff base ligands, L′, were always found as monomeric species in a pseudo-tetrahedral coordination around the Zn(II) atom (Figure 2) [19,20,21,22], independently from the presence of Lewis bases. In other terms, in the absence of sterical constraints the formation of tetrahedral structures is favored.

This issue was recently explored theoretically by analyzing a series of complexes varying the bridging diimine [23,24]. It was found that the diimine bridge in ZnL model monomers forces the ligand in a distorted planar coordination with a large atomic charge on the zinc atom, leading to Lewis acidic, electronically and coordinatively, unsaturated species.

Therefore, it turns out that in absence of Lewis bases these complexes are generally stabilized through intermolecular Zn⋯O axial interactions involving the phenolic oxygen atoms of the ligand framework, leading to penta-coordinated square-pyramidal Zn(II) geometries [15]. Unless there is a sterical hindrance by bulky substituents in the salicylidene rings [25], or the Zn(II) metalation involves shape persistent L macrocycles [26], the mutual Zn⋯O interaction leads to the formation of dimers [27,28,29,30,31], while larger oligomeric aggregates, including mesomorphic [32,33,34,35] and nanostructured [36,37,38,39,40,41,42,43,44,45,46] species are obtained in the other cases. On the other hand, in the presence of monotopic Lewis bases ZnL complexes form monomeric adducts [28,30,47,48,49,50,51,52] with the base axially coordinated to the metal (Scheme 1).

Alternatively, a self-assembly of ZnL complexes can be achieved with an appropriate donor substituent on the ligand frameworks [53,54,55], or by reaction of ZnL complexes with various ditopic bipyridine ligands [56]. Moreover, a different aggregation behavior occurs in the presence of anionic species. Kleij et al. first demonstrated the anion templated formation of supramolecular assemblies [57]. For example, the acetate anion can lead both to 1:1 and 2:1 bridged assemblies controlled by variation of the concentration and stoichiometry of the two components (Figure 3). Otherwise, unsymmetrical ZnL complexes bearing on one side an alkyl ammonium bromide as a Lewis base, form dimeric aggregates [58] in which each molecular unit mutually interacts with another unit through intermolecular Zn⋯Br^−^ interactions (Figure 3), while related symmetrical species lead to the formation of branched nanostructures [59].

In all instances, the formation of aggregates, (ZnL)_n_, or adducts leads to species with profound differences in their spectroscopic properties, beyond their structural variation, which will be described in the next paragraphs.

## 3. Spectroscopic Properties of ZnL Complexes in Solution

### 3.1. ^1^H NMR Studies

Di Bella, et al. were involved in several systematic studies on ZnL complexes (**1**–**6**, Figure 4) in solution in the presence/absence of coordinating solvents [27,60,61,62,63]. To ensure a suitable solubility of ZnL complexes in both kinds of solvents, often it was necessary to synthesize amphiphilic species containing alkyl substituents in the salicylidene rings. The synthesis of ZnL complexes was generally achieved in polar coordinating solvents. Under these conditions, if the solvent possesses a sufficient Lewis basicity, monomeric adducts are formed. However, if the purified solid complexes are heated under vacuum, the solvent is desorbed and a ground solid is obtained. Dissolving the ground solid either in coordinating or non-coordinating solvents, very different spectroscopic properties, in relation to the nature of the bridging diimine, are observed.

Complexes **1**–**6** in solution of the DMSO-*d*_6_ coordinating solvent are well characterized as monomeric adducts with the solvent axially coordinated. On switching to non-coordinating solvents, such as CDCl_3_ or CD_2_Cl_2_, a significant change of ^1^H NMR spectra was always observed (Figure 5). Except for complex **2**, a general up-field shift for the aromatic as well as for imine protons is found (Table 1). This is particularly evident for H_1_ (up to 0.41 ppm), H_3_ (up to 0.63 ppm), and H_4_ (up to 0.55 ppm) signals. These observed up-field shifts suggest that the involved hydrogens lie under the shielding zone of the π electrons of a conjugated system, and are consistent with the presence of aggregate species in solution. Moreover, ^1^H NMR signals in non-coordinating solvents are often concentration dependent. Thus, in the case of complexes **1**, **3**, and **4** a progressive peak broadening of above involved signals was observed. This is typical of the presence of larger aggregate species.

These results were corroborated by ^1^H diffusion ordered spectroscopy (DOSY) NMR studies which always confirmed the presence of dimers or larger aggregates species of **1**–**6** in solution of non-coordinated solvents, in contrast to the existence of monomeric adducts in DMSO-*d*_6_ solution.

To obtain a sufficient solubility both in non-coordinating and coordinating solvents, various derivatives, changing the length of the alkoxy chain, were synthesized. However, results shown that the alkyl side chains length plays a very minor role in the aggregation properties of such species [60,61,62]. Interesting enough is the ^1^H NMR behavior observed for the complex derived from *cis*-1,2-diaminocyclohexane **2** in CDCl_3_ solution [27]. In fact, H_2_, H_3_, H_4_, and –OCH_3_ signals are split into two signals with the same intensity, consistent with a single dimeric species having an asymmetric structure, further supported by DOSY NMR, as also observed in the solid state by X-ray crystal structure analysis [27].

Trinuclear ZnL macrocyclic complexes **7** (Figure 6) in two different solvents were studies by MacLachlan et al. [64]. The ^1^H NMR spectroscopic analysis in THF-*d*_8_ showed the presence of sharp signals which were assigned to the macrocycle. In contrast, broadened signals were observed in mixture of CD_2_Cl_2_/THF-*d*_8_ solvents indicating the formation of large aggregates. Analogous results were obtained by Kleij et al. [65] on the tetranuclear macrocycle **8** (Figure 6).

From these studies emerges that different species in solution, in relation to the nature of solvent, are formed. While in non-coordinating solvents dimeric/oligomeric species are present, in coordinating solvents monomeric adducts with solvent coordinated are always found.

### 3.2. ^1^H NMR Studies of Chiral Complexes

Some chiral ZnL complexes (**9**–**13**) were also investigated (Figure 7). In particular, complexes derived from the enantiopure chiral (1*S*,2*S*)-(+)- or (1*R*,2*R*)-(‒)-*trans*-1,2-diaminocyclohexane and (1*R*,2*R*)-*trans*-1,2-cyclopentanediamine were synthesized and studied in solution [66,67,68], in comparison to previous achiral complexes [27,60,61,62,63].

Although the ^1^H NMR spectrum of **9** in solution of the coordinating DMSO-*d*_6_ solvent suggests the presence of the usual monomeric adduct, in CDCl_3_ behaves very differently with respect to the other previously investigated complexes, including the **2**
*cis*- analogue [27,66]. In fact, the ^1^H NMR spectrum indicates the existence of three species in solution, **9A**–**C**, with a pronounced concentration dependence. While **9B** and **9C** can be associated to dimeric species, **9A** can be related to an oligomeric species involving ca. 20 monomeric units, as resulted from ^1^H NMR DOSY studies (Figure 8).

More interesting, after standing or by heating all species are converted into **9C**, which is very stable and hardly disaggregable by addition of DMSO, indicating a presumable different aggregation mode. In fact, a stronger up-field shift of the imine hydrogens, than that observed for the dimeric aggregates involving a penta-coordination for the Zn(II) ion, is found for this species [66]. This chemical shift is comparable to that found for **12** [66], and for the chiral *trans*-1,2-diaminocyclohexane-(pyrrol-2-ylmethyleneamine) (**14**) derivative [69], both complexes characterized by X-ray analysis as a double-stranded dinuclear helicate structure with a tetrahedral coordination around the Zn(II) ion [69,70]. For this family of complexes, it was established that the chemical shift of the imine hydrogens is diagnostic to predict their aggregation mode (Table 2). Thus, for the aggregate species **9C** was proposed an analogous double-helicate Zn_2_L_2_ structure, as a consequence of the defined stereochemistry of the chiral *trans*-1,2-diaminocyclohexane chelate bridge (Figure 9).

The 4-diethylamino, instead of the methoxy, substituent on the salicylidene rings allowed to obtain a more soluble complex, **11**, which presents further interesting properties. In particular, the greater solubility of the complex enables a higher degree of aggregation in solution, up to ca. 27 monomeric units [67]. For these aggregates, obtained in freshly-prepared solutions of non-coordinating solvents, a helical oligomeric structure was proposed, (ZnL)_n_, with a tetrahedral coordination geometry around the metal center. By standing or heating these aggregates evolve to a more stable dinuclear double-helicate Zn_2_L_2_ species, **11-h** (Figure 10).

The *trans*-1,2-cyclopentanediamine **13** derivative [68], in comparison with the *trans*-1,2-diaminocyclohexane **9** analogue [66], showed unexpected behavior. In fact, experimental data of **13** in DMSO solution suggest the existence of two species, the monomeric adduct with the solvent and a dinuclear double-helicate dimer. The formation of monomeric adducts occurs only when dissolving the complex in a stronger Lewis base, such as pyridine. In the non-coordinating chloroform solvent, **13** shows the presence of a single defined dimeric species, in every experimental condition, which does not deaggregate even with the addition of pyridine in large stoichiometric excess. Again, the chemical shift of the imine hydrogens is consistent with a dinuclear double-helicate Zn_2_L_2_ species (Table 2). This study demonstrated that even a slight change in the structure of the cycloaliphatic ring of the *trans*-1,2-diamine bridge plays a major role in the aggregation properties.

In summary, the defined *trans*- stereochemistry of the cyclohexane- or cyclopentane-1,2-diamine, in comparison to the non-chiral species, leads to a different aggregation mode of these chiral ZnL complexes in non-coordinating solvents, with formation of dinuclear double-helicate Zn_2_L_2_ structures, having a tetrahedral coordination around the Zn(II) atoms. As a consequence, such complexes are thermodynamically more stable than related pentacoordinated species.

### 3.3. Optical Absorption and Fluorescence Studies

Optical absorption spectra of ZnL complexes in the visible region are strictly related to the nature of the bridging diimine, involving molecular orbitals delocalized over the entire molecules by π→π* transitions [71,72], being those derived from conjugated species certainly more interesting. Thus, in the case of complexes **3**–**6** UV/Vis absorption spectra in non-coordinating solvents indicate the existence of aggregate species, as they are generally characterized by structureless features with larger bandwidths, compared to those recorded in coordinating solvents [58,61,62,63,71]. On changing to coordinating solvents, e.g., DMSO or THF, the longer wavelength band becomes more intense and red-shifted, suggesting an *H*-type coupling between the salicylidene groups of each unit in the aggregate species. Thus, in the case of the Zn(salmal) complex derivative, **3**, a naked eye color change, from red-orange to fuchsia, on switching from DCM to THF solutions is observed [62,63] (Figure 11). Therefore, the formation of the monomeric adducts involves relevant changes of optical spectroscopic properties. In contrast, for the non-conjugated bridging diimine derivatives **1**, **2**, and the chiral **9**–**11**, and **13** complexes, are observed minor variations of the UV/Vis absorption spectra on changing the coordinating nature of the solvent, indicative of weak interligand interactions between each unit in aggregate species of these complexes [27,60,66,67,68].

Conversely, the chiral *trans*- **9** and **11** derivatives exhibit a further interesting behavior. While UV/Vis spectra of fresh prepared solutions in both coordinating/non-coordinating solvents are very similar one each other, spectra of **9C** and **11-h**, achieved upon heating/standing chloroform solutions of **9** and **11**, are characterized by the appearance of a new, intense absorption band at longer wavelengths [66,67]. This is consistent with a change of their structure, from the oligomeric (ZnL)_n_ to the double-helicate Zn_2_L_2_ dimer, the latter being characterized by stronger interligand interactions. An analogous behavior is observed for **13** [68]. In fact, while the monomeric adduct with pyridine shows an absorption band centered at λ = 348 nm, the double-helicate Zn_2_L_2_ dimer in chloroform behaves very differently, with the presence of a new, more intense absorption band at λ = 374 nm.

Circular dichroism (CD) studies of the chiral complexes **9**–**11** in CHCl_3_ solution allowed to further probe on the aggregation properties changing from the oligomeric (ZnL)_n_ to the helicate Zn_2_L_2_ species. In particular, on passing from (***R***)-**9**, (***S***)-**10**, or (***R***)-**11** to (***R***)-**9C**, (***S***)-**10C** or (***R***)-**11**-**h** enhanced bisignate CD signals are observed, consistent with the formation of a double-helicate Zn_2_L_2_ structure (Figure 12) [66,67]. On the other hand, the CD spectrum of **13** in CHCl_3_ solution exhibits strong bisignate signals, according with the existence of a single defined dinuclear double-helicate species [68].

Since the d^10^ electron configuration of the Zn(II) ion, the fluorescence of ZnL complexes in solution can essentially be related to that of the L^2−^ ion, whose associated transitions are originating from singlet ligand-centered excited states [73]. Actually, ZnL complexes are generally emissive species, whose fluorescence is modulated by the structure of the bridging diimine [74]. Derivatives from Zn(salphen) [61,74,75,76,77] and Zn(salmal) [28,50,58,59,62,63,78,79] are those most investigated. Interesting, these complexes are characterized by very different fluorescent properties on switching from the aggregate species to the monomeric adducts. For example, while DCM solutions of **3** exhibit a moderate orange fluorescence emission (λ_em_ = 597 nm; *Φ* = 0.07), a substantial enhancement of the fluorescence (λ_em_ = 593 nm; *Φ* = 0.24) is observed in THF [62] (Figure 11). An analogous, even if less evident, effect was found for complexes **4**–**6**. Therefore, aggregation of ZnL complexes is accompanied by partial quenching of their fluorescence because of intermolecular Zn⋯O interactions.

Given the interesting photophysical characteristics of ZnL complexes, various studies have been done mainly involving derivatives of Zn(salmal) and Zn(salphen), in relation to their fluorescence in self-assembled mono- [80,81] or multilayers [82], in mesomorphic systems [32,33,34,35,83,84], for potential applications as dye sensitized solar cells [47,79,85], and with mechanochromic [86] and organogelation [46,86] properties.

### 3.4. Deaggregation Studies

Since the aggregation or the formation of monomeric adducts of ZnL complexes in solution depends upon coordinating properties of the involved solvent, the following questions can be raised. What happen to an aggregate ZnL species in a non-coordinating solvent upon addition of a Lewis base? If a deaggregation occurs, what spectroscopic changes are to be expected?

The deaggregation of supramolecular assemblies of the conjugated trinuclear macrocycle complexes **7** or of polymeric species containing the ZnL unit **15** (Figure 13) was first investigated by MacLachlan et al. [64,87]. It was found that the addition of a strong Lewis base, such as pyridine, to a DCM solution of the aggregate macrocycles **7** or the polymeric species **15** causes deaggregation and results in a large enhancement of their fluorescence.

A series of crystallographically characterized (non)symmetrical Zn(salophen) dimers having *tert*-butyl or other substituents in the salicylidene rings (**16**, Figure 13) were investigated by Kleij et al. [29]. Again, starting from toluene solutions of these dimeric species, addition of a large excess of pyridine leads to significant optical absorption spectral changes with formation of monomers.

Systematic studies on deaggregation properties of complexes **1**–**6** in solution of CH_2_Cl_2_ or CHCl_3_ were carried out by Di Bella et al. [27,60,61,62,63]. In all cases, the addition of a Lewis base, such as DMSO or pyridine, always leads to deaggregation of the aggregate species with formation of the monomeric adduct, as demonstrated by ^1^H NMR and DOSY studies. Moreover, relevant changes of UV/Vis and fluorescence spectra were found. In particular, deaggregation of complex **3** involves a relevant change of the optical absorption spectrum and a dramatic enhancement of the fluorescence (Figure 14) [62,63]. Moreover, deaggregation of **3** leads to an unprecedented switching-on of the second-order nonlinear optical properties, from a *μβ*_1.907_ value nearly zero for the aggregate to a *μβ*_1.907_ = −2070 × 10^−48^ esu upon deaggregation [88].

As the Lewis acidic character of ZnL complexes involves different aggregation/deaggregation properties which, in turn, are reflected in their sensing properties (see next paragraph), it is crucial the evaluation of the Lewis acidity on varying the nature of the bridging diimine. The experimental and calculated binding constants (Table 3) for the formation of adducts with pyridine (Equation (1)) upon deaggregation of (ZnL)_2_ dimers were associated to the relative Lewis acidity scale within the series of complexes **1**–**6** [23].
(ZnL)_2_ + 2py → 2ZnL py(1)

The Lewis acidic character varies with the nature of the diimine bridge, being stronger in the case of complexes of conjugated diimines (Table 3). While the quasi-planarity of ZnL monomers containing conjugated diimine bridges is an unfavorable aspect in the dimerization process, it leads to the stabilization of ZnL·py adducts in a square-pyramidal coordination [23]. These results are relevant for sensing and catalytic properties of these complexes.

As expected, in the case of chiral complexes **9**–**13** the formation of tetrahedral double-helicate Zn_2_L_2_ structures leads to species with a weak Lewis acidic character, hardly disaggregable [66,67]. In particular, deaggregation of **13** occurs only when dissolving the complex in the strong Lewis base pyridine [68].

The addition of a Lewis base to ZnL aggregates affording formation of monomeric adducts does not seem to be the unique way for deaggregation. Actually, deaggregation of a derived Zn(salamal) complex was demonstrated by addition of Hg^2+^ ions [89].

## 4. Sensing Properties of ZnL Complexes

The relevant changes of spectroscopic properties observed upon deaggregation of ZnL complexes and their different Lewis acidic character on changing the bridging diimine opened to various studies as possible sensors for Lewis bases.

Dalla Cort et al. first investigated the binding of tertiary amines to substituted Zn(salphen) complexes (**17** and **18**, Figure 15) in CHCl_3_ by spectrophotometric titrations [90]. They found a dramatic influence of steric effects leading to a selectivity among the series of amines investigated, with association constants >10^6^ in the case of quinuclidine.

Analogues studies, through UV/Vis and NMR spectroscopy, by Kleij et al. involved complex **19** and the related dinuclear species **20** (Figure 14) as efficient materials for binding pyridine-based alkaloid derivatives (binding constants > 10^5^) [91]. The X-ray structure of the monomeric adducts with nicotine was also determined. The complex **20** can be also reversibly metalated by Zn^2+^ ions in wet CHCl_3_ in the presence of suitable *N*-donor ligands. The method allowed the colorimetric discrimination between alkaloid species containing quinoline nuclei [92].

Complex **3** in dichloromethane solution was used as sensitive Lewis acid for the detection of a series of aliphatic amines by means of the fluorescence enhancement upon amine coordination to the metal center [93]. A selectivity and nanomolar sensitivity was found for primary and alicyclic amines. By means of the derived binding constants a relative Lewis basicity was established for acyclic amines, primary > secondary > tertiary, with an inverted order, tertiary > secondary ≈ primary (acyclic), for alicyclic amines. The inverted order of basicity was related to the steric hindrance at the nitrogen atom [93]. Moreover, **3** was also involved as chromogenic and fluorogenic probe for detection of some classes of alkaloids. A high selectivity, in the micromolar range, and sensitivity was found for pyridine-based and cinchona alkaloids (Figure 16) [94].

The relative binding constants for formation of Lewis base adducts were used to rank the Lewis basicity of involved species [95,96]. Thus, using complex **3** as “real world” reference for Lewis acidic species, instead of the simpler reference molecules such as BF_3_ or SbCl_5_, it was possible to build a scale of Lewis basicity in dichloromethane for a large variety of bases, involving neutral amines, charged anions, and non-protogenic solvents. All involved anions are characterized by a strong Lewis basicity, rivalling or exceeding that of the stronger neutral bases, such as primary amines or pyridine (Figure 17). In turn, for neutral species the Lewis basicity seems governed by the steric hindrance at the donor atom. These results are expected to be relevant in the organic synthesis and catalysis, given the nature of Lewis acidic species commonly involved.

The complex **19** was also involved for the fluorescent discrimination of nitroaromatics. Knapp et al. first demonstrated that this complex undergoes moderately efficient fluorescence quenching in the solution phase by electron transfer with nitroaromatics and 2,3-dimethyl-2,3-dinitrobutane [97]. By using a series of Zn(salphen) complexes with various substituent as in **16**, was explored a fluorescence-based sensor array which accurately discriminated nitroaromatics and explosive mimics [98].

Complexes **17** and **18** were also involved in an extensive investigation for the binding properties toward inorganic phosphates and adenosine nucleotides in ethanol by means of different spectroscopic techniques [99]. These complexes resulted good receptors to both series of phosphates, and efficient fluorescent chemosensors for adenosine nucleotides (Figure 18).

A series of Zn(salphen) complexes substituted in the para position to the phenolic oxygens were analogously used for the molecular recognition of anions in ethanol. The results showed the possibility of tuning the selectivity of recognition through the appropriate choice of substituents [100].

As far as the recognition of anions in aqueous solutions, a Zn(II) complex derivative from pyridoxal with ethylenediamine (**21**, Figure 19) was found to be selective for the fluorescent detection of ADP and ATP [101]. This is consequence of the complementary coordination of the phosphate groups to the metal center, corroborated by hydrogen bonding of the nucleotide with the coordinated ligand. Moreover, a Zn(salphen)-bile-acid conjugate in an aqueous solution of CTABr micelles was found to bind phosphate, analogously to the binding of anions of the complex in DMSO solution [102].

Complexes **19** and **20** were also investigated for sensing anions. These complexes in acetone solution give an anion-specific reaction with dihydrogen phosphate because of a proton transfer to the chromophore with consequent demetallation, accompanied by appreciable color change [103]. Using the same strategy and the simpler substituted Zn(salen) derivative, it was demonstrated the detection of di- and triphosphates in water, upon demetallation and formation of the fluorescent salicylaldehyde [104].

As the demetallation process is a consequence of a proton transfer, it is strictly related to the pH of the medium. Thus, using a substituted Zn(salmal) derivative in CH_3_CN it was demonstrated that this complex exhibits sensitive and selective off‒on‒on′‒off pH fluorescence response, by three different reaction mechanisms including demetallation, protonation, and hydrolysis reaction [105].

The use of chiral substituents or chiral 1,2-diamines in ZnL complexes is expected to discriminate chiral Lewis bases. Thus, a new water soluble Zn(salphen) complex with appended d-glucose was proven to give an unexpected chiral discrimination toward six aminoacids. The latter, bind to the complex via two interactions, coordination of the carboxylate to the Zn(II) atom and hydrogen bonds between the ammonium group of the aminoacid and the alcoholic groups of the d-glucose moiety [106]. Analogously, using chiral Zn(salen) complexes with a (*R*,*R*)-configuration in the bridge, it was reported a recognition toward some (*R*) or (*S*) amines [107]. On the other hand, a calix[4]arene host having two Zn(salphen) derivatives in the presence of a chiral diamine guest self-assembles into a capsular complex with a chirality transfer from the diamine to the calix–salphen host [108]. Likewise, using di- and trinuclear Zn(salphen)-based host complexes and chiral guest ditopic species it was demonstrated a supramolecular chirality induction to the obtained adducts [109].

The good fluorescent characteristics, the lack of redox activity of Zn(II) and its large abundance in living organisms, rends ZnL complexes suitable candidates for biological studies [18]. Zhang et al. investigated a series of luminescent Zn(salmal) derivatives having lipophilic or cationic substituents in the salicylidene rings. These complexes exhibited low cytotoxicity to living cells and subcellular selectivity, ideal as optical probes in living cell imaging [110]. Moreover, they demonstrated that intermolecular Zn⋯O interactions in these substituted Zn(salmal) derivatives play a relevant role in determining their cellular uptake pathway and subcellular distribution [111] (Figure 20).

Complexes **17** and **18** were also involved by Rodríguez et al. for their biological activity. It was observed that these complexes show cellular uptake with a non-cytotoxic behavior [112]. Moreover, the same authors studied the interaction of a series of Zn(salphen) derivatives with free plasmid DNA. Variation in the electronic properties of substituents at the 5,5′ positions of the salicylidene rings seems to modulate the strength of such interactions [113].

More recently, the aggregation/deaggregation properties of ZnL complexes were explored in the solid phase. In particular, the ground solids of **3** and **4**, upon an irreversible thermal structural phase transition, when exposed to vapors of a Lewis base, such as a volatile amine, exhibit a marked change of both the optical absorption spectra [114] and the resistivity [115,116], because of chemisorption of the Lewis base with the formation of adducts. The process is reversible, as the chemisorbed material is easily restored by thermal treatment, inducing desorption of the Lewis base, and is reproducible in successive cycles of exposure/restoring. Therefore, these complexes are promising molecular materials as sensors for volatile organic Lewis bases. Moreover, a supramolecular gel based on a Zn(II)–salophen bis-dipeptide derivative was found to be responsive to the presence of anions [117].

## 5. Conclusions

Bis(salicylaldiminato)Zn(II) Schiff-base complexes from 1,2-diamines have recently been studied for their interesting aggregation properties. Since monomeric ZnL having a quasi-planar coordination around the Zn(II) atom would be coordinatively unsaturated, these Lewis acidic complexes are stabilized by axial coordination with an appropriate Lewis base. This leads to square-pyramidal pentacoordinated structures, with formation of varied molecular architectures. Compared to common molecular aggregates, this aggregation “via coordination chemistry” allows for a better control of the resulting supramolecular structure/nanostructure by the appropriate choice of the neutral/charged, monotopic/multitopic Lewis bases, including Lewis bases belonging to the ligand or to its substituents. However, in the case of chiral ZnL complexes from 1,2-diamines with a defined *trans*- stereochemistry a different aggregation mode occurs, with formation of dinuclear double-helicate Zn_2_L_2_ structures having a tetrahedral coordination around the Zn(II) atoms.

The Lewis acidic character of ZnL complexes varies with the nature of the diimine bridge, being stronger in the case of complexes of conjugated diimines. This is reflected in their different aggregation/deaggregation and sensing properties. Deaggregation of these structures in solution occurs by addition of an appropriate Lewis base, with formation of monomeric adducts having very different spectroscopic characteristics. This represented the starting point for exploiting these complexes as chromogenic and fluorogenic chemosensors of Lewis bases, including cell imaging. Thus, a variety of Lewis bases, involving charged and neutral species, have been investigated in sensing studies, mainly using complexes having a Zn(salphen) or Zn(salmal) structure. Moreover, ZnL complexes resulted to be selective and sensitive to the Lewis basicity of the involved species. Finally, these complexes have recently emerged as vapochromic and chemiresistive materials. In summary, thanks to their Lewis acidic character, ZnL Schiff-base complexes are classical coordination compounds for modern applications.

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
