# Peer review of "On the Aggregation and Sensing Properties of Zinc(II) Schiff-Base Complexes of Salen-Type Ligands"

_molecules, 2019, doi:10.3390/molecules24132514_

Round 1
Reviewer 1 Report
This is a very comprehensive review and I should have asked for more time to thoroughly go over this work in order to provide more in-depth comments and suggestions. However, after an initial reading this is some interesting chemistry here and the paper is relatively well written with some minor grammatical and English issues that could hopefully be cleared up in the editorial stages. The literature is thoroughly covered and the references are presented in a clear and correct fashion. My only comment, and it is a minor one and offered solely in a constructive manner, is that in some of the Figures atom fonts vary between Arial and Times New Roman (please see Figure 1 where the zinc complex is in Arial but the R group in the backbone and in the three examples to the right of the figure are in Times New Roman. It is not a big deal but I think it would look much better if all were the same font (either one is good). Overall, an enjoyable read and I look forward to reading more from the authors in the near future.
Author Response
According to the reviewer’s suggestion we have redrawn all Figures involving ChemDraw structures, using for atoms labels the Times New Roman font.
Reviewer 2 Report
This review reports on the aggregation ans sensing properties of Zn salen-type compounds.
It is well organized and seems to be an interesting tool for scientists working in this field.
I would only ask the authors to introduce the assignment of absorption and emission bands / transitions in the optical properties section since it will enrich the scientific discussion of changes on recorded bands (shifts and colors).
Author Response
According to the reviewer’s suggestion in the paragraph 3.3. we have added some sentences about a general assignment of absorption and emission band transitions of ZnL complexes. Moreover, a new reference, Ref. 73, has been added in the revised version of the manuscript.